# Dysregulated Alternative Splicing in Breast Cancer Subtypes of RIF1 and Other Transcripts

**DOI:** 10.3390/ijms26157308

**Published:** 2025-07-29

**Authors:** Emma Parker, Laura Akintche, Alexandra Pyatnitskaya, Shin-ichiro Hiraga, Anne D. Donaldson

**Affiliations:** Chromosome & Cellular Dynamics Section, Institute of Medical Sciences, University of Aberdeen, Aberdeen AB25 2ZD, UK

**Keywords:** breast cancer, RNA splicing, TCGA

## Abstract

Genome instability is a hallmark of cancer, often driven by mutations and altered expression of genome maintenance factors involved in DNA replication and repair. Rap1 Interacting Factor 1 (RIF1) plays a crucial role in genome stability and is implicated in cancer pathogenesis. Cells express two RIF1 splice variants, RIF1-Long and RIF1-Short, which differ in their ability to protect cells from DNA replication stress. Here, we investigate differential expression and splicing of RIF1 in cancer cell lines following replication stress and in patients using matched normal and tumour data from The Cancer Genome Atlas (TCGA). Overall *RIF1* expression is altered in several cancer types, with increased transcript levels in colon and lung cancers. *RIF1* also exhibits distinct splicing patterns, particularly in specific breast cancer subtypes. In Luminal A (LumA), Luminal B (LumB), and HER2-enriched breast cancers (HER2E), *RIF1* Exon 31 tends to be excluded, favouring RIF1-Short expression and correlating with poorer clinical outcomes. These breast cancer subtypes also tend to exclude other short exons, suggesting length-dependent splicing dysregulation. Basal breast cancer also shows exon exclusion, but unlike other subtypes, it shows no short-exon bias. Surprisingly, however, in basal breast cancer, *RIF1* Exon 31 is not consistently excluded, which may impact prognosis since RIF1-Long protects against replication stress.

## 1. Introduction

Genome instability is closely intertwined with carcinogenesis, and tumours almost invariably show mutation or altered expression of genome maintenance factors that regulate DNA replication and repair [1,2]. Consistently, prolonged DNA replication stress and associated DNA damage are important contributors to tumorigenesis. Cancer cells often experience elevated replication stress induced by DNA repair protein mutations, checkpoint inactivation, and rapid cell division. Many cancer therapies exploit this vulnerability, inducing irreparable DNA damage to eliminate cells with compromised genome maintenance pathways [3,4].

Mammalian RIF1 (Rap1 Interacting Factor 1) is a multifunctional protein with numerous roles in genome stability. RIF1 expression has been reported as upregulated across different tumour types, including breast, cervical, ovarian, and non-small cell lung cancers [5,6,7,8]. This upregulation has been correlated with enhanced tumour growth, migration, and cell survival following radio- and chemotherapy. The elevated expression of RIF1 within certain tumours may affect their vulnerability to specific DNA-damaging treatments, offering a potential avenue for targeted therapeutic strategies [9]. It is therefore important to understand RIF1 expression changes in cancer.

RIF1 serves to regulate both DNA repair and replication. In its DNA repair role, RIF1 is recruited to double-strand breaks (DSB) through interaction with phospho-53BP1 to direct repair to the non-homologous end-joining pathway and suppress homology-based repair [10,11]. RIF1 is furthermore crucial for cell survival after replication stress, indicative of its involvement in regulating DNA replication and events at challenged replication forks [12]. RIF1 protects the nascent DNA at stalled forks [13,14], promoting fork restart and suppressing fork collapse and associated DSB formation. However, its role in nascent DNA protection only partially accounts for the role of RIF1 in cellular recovery from replication stress [15].

Human *RIF1* undergoes alternative splicing, yielding two protein isoforms, called RIF1-Long (RIF1-L) and RIF1-Short (RIF1-S). The RIF1-S isoform lacks 26 amino acids near the protein C-terminus due to exclusion of Exon 31 (Figure 1A). Although RIF1-L is specified as the canonical form by Uniprot, many studies aiming to dissect RIF1 functions have examined only the RIF1-S form [11,16]. The two RIF1 isoforms are, however, not equivalent in protecting cells from drugs that induce replication stress. Specifically, cells containing only RIF1-S exhibit sensitivity to replication stress-inducing drugs, while cells that contain only RIF1-L show levels of resistance to replication stress that are similar to normal cells [15]. RIF1-L, therefore, appears capable of mediating replication stress resistance, while RIF1-S is not. RIF1-L Exon 31 contains a BRCT interaction motif (i.e., an SPxF motif) that was recently shown to mediate interaction with the BRCT domain of BRCA1 under extended replication stress conditions, in an interaction that promotes recombination-mediated replication fork recovery [17]. Therefore, RIF1-L is needed to direct specific recovery pathways at sites of interrupted DNA replication, contributing to the function of RIF1 in mediating resistance to replication stress.

Aberrant RNA splicing is increasingly recognised as a hallmark of cancer that contributes to tumour progression. Mutations in components of the splicing machinery (e.g., SF3B1) or splicing regulators (e.g., HnRNPs) have been observed across many tumour types and lead to widespread alterations in exon usage [18]. This splicing dysregulation can produce protein isoforms with altered functionality, promoting cancer cell survival [19]. Alternative splicing is also affected by RNA polymerase (RNAPII) elongation kinetics, thought to be dysregulated in cancer due to oncogene activation and resulting elevation in transcription. Slow RNAPII elongation favours exon inclusion by allowing more time for recognition of weak splice sites [20]. Importantly, splicing dysregulation has been linked to the DNA damage response and replication stress tolerance, as several DNA repair factors are subject to alternative splicing [21].

An early publication mentioned that transcripts encoding RIF1-S were found to be more abundant in cancer cell lines [22]. However, whether the two RIF1 isoforms show differential expression in cancer patients has remained unclear. Given the different functionality of RIF1-L and RIF1-S, it is important to understand the expression characteristics and isoform-specific roles of RIF1 in cancer, as such information will help elucidate the role of RIF1 in protecting cancer cells from replication stress. Based on the clinical importance of chemotherapeutic agents that interfere with the DNA replication process, investigating RIF1 isoform expression may also provide insights into how tumour cells respond to replication-inhibiting therapeutic drugs.

**Figure 1 ijms-26-07308-f001:**
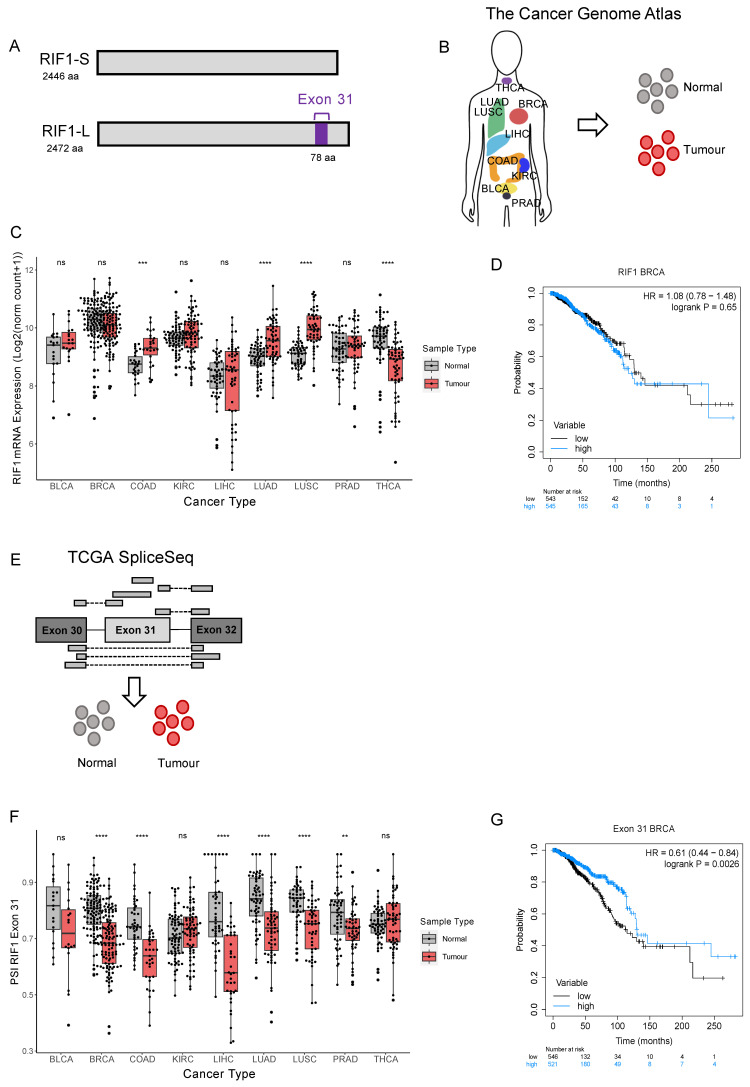
*RIF1-L* proportion is decreased in several tumour types and in breast cancer correlates with poor prognosis. (**A**) Schematic diagram of RIF1-S and RIF1-L, where only RIF1-L contains Exon 31. (**B**) Illustration showing tumour types included in The Cancer Genome Atlas repository, from which *RIF1* gene expression data were downloaded for tumour and matched normal samples (adapted from Corces et al., (2018) [23]). (**C**) *RIF1* mRNA expression (as Log2 (norm count + 1)) in tumour (red) and matched normal samples (grey) in different cancer types. Statistical comparison was carried out using paired Wilcoxon signed-rank test, *** *p* < 0.001, **** *p* < 0.0001, ns: non-significant. (**D**) Kaplan–Meier survival curve for breast cancer patients grouped into ‘high’ (blue) and ‘low’ (black) expression by median expression of *RIF1*. HR = hazard ratio. (**E**) Illustration showing how PSI (percent spliced-in) values are calculated. Reads shown above the *RIF1* Exon structure cartoon contain Exon 31 and encode RIF1-L, while reads shown below do not contain Exon 31 and encode RIF1-S. PSI values were downloaded for tumour and matched normal samples. (**F**) PSI values for *RIF1* Exon 31 in tumour (red) and matched normal samples (grey) in different cancer types. Statistical comparison was carried out using paired Wilcoxon signed-rank test, ** *p* < 0.01, **** *p* < 0.0001, ns: non-significant. (**G**) Kaplan–Meier survival curve for breast cancer patients grouped into ‘high’ (blue) and ‘low’ (black) by median PSI Exon 31.

Here, we use data obtained from matched normal and tumour tissue samples from individual patients to show that the transcript encoding the RIF1-S splice variant increases in abundance in cancers, including breast cancer. We find, moreover, that receptor-positive breast cancers have a generally higher propensity to exclude short exons, including Exon 31 of *RIF1*. Basal breast cancer also shows a shift towards exon exclusion but differs from LumA, LumB, and HER2-enriched subtypes in that the long and short exons are equally likely to be excluded. In basal cancer, however, *RIF1* Exon 31 does not follow this typical pattern but instead continues to be included at a similar rate as in normal tissue. Since RIF1-L specifically is required to protect cells from replication stress, our results may relate to the worse prognosis associated with basal breast cancer compared to the other subtypes.

## 2. Results

### 2.1. RIF1 Expression and Splicing Are Altered in Cancer in Some Tumour Types

RIF1 overexpression has been reported in several cancer types, including breast, ovarian, cervical, and non-small cell lung cancer (NSCLC) [5,6,7,8]. These findings have been based on analyses of fixed tissue samples and RNA-sequencing data held in various online repositories. Data from The Cancer Genome Atlas (TCGA) have been used previously to investigate *RIF1* expression in NSCLC [24], the most common lung cancer subtype. Here, we use *RIF1* mRNA expression data from TCGA [25] to investigate total *RIF1* expression in a larger number of cancer types, selected based on availability of matched normal samples (Figure 1B). We performed paired Wilcoxon signed-rank tests to compare *RIF1* mRNA expression between normal and tumour samples (Figure 1C). We observed an increased total *RIF1* mRNA in colon (COAD), lung adenocarcinoma (LUAD), and lung squamous (LUSC) cancers and decreased expression in thyroid cancer (THCA) compared to matched normal tissues (Figure 1C). Other cancer types analysed show no significant difference in *RIF1* expression, including bladder (BLCA), kidney (KIRC), liver (LIHC), prostate (PRAD), and, interestingly, breast cancer (BRCA), despite a previous report suggesting RIF1 is highly expressed in malignant breast tumour sections [7].

To understand if changes in *RIF1* expression are predictive of patient survival, we performed Kaplan–Meier analyses for all cancer types analysed in Figure 1C, grouping patients based on median *RIF1* mRNA expression levels (Appendix A). We found that high *RIF1* expression is associated with improved patient survival in kidney cancer (Appendix AD). However, in other cancer types, including breast cancer, we did not find a significant correlation between *RIF1* expression and survival in breast cancer (Figure 1D and Appendix A).

To investigate the role of RIF1 splice variants in cancer, we used transcript analysis data held by TCGA SpliceSeq to analyse inclusion of Exon 31 in *RIF1* transcripts from different tumour types [26]. This includes ‘percent spliced-in’ (PSI) values for all observed exon skips, calculated using RNA-sequencing data obtained from TCGA (Figure 1E). PSI values indicate the proportion of normalised read counts in a sample that contain a given exon (i.e., a PSI of 0.5 indicates half of the reads covering that locus in the sample contain the Exon 31 and half do not). Figure 1F shows PSI values for *RIF1* Exon 31 compared to matched normal samples from the same patients using paired Wilcoxon signed-rank tests. In breast, colon, liver, lung (adenocarcinoma and squamous), and prostate cancers, PSI values for Exon 31 were significantly reduced in tumours compared to normal tissues, indicating reduced *RIF1-L* transcript in these cancer types (Figure 1F). For kidney cancer, there were no significant alterations in PSI Exon 31. No significant difference in PSI Exon 31 was also observed for bladder and thyroid cancers; however, tumour samples from these cancer types did show a wider range of PSI values compared to the matched normals. This indicates changes in *RIF1* splice variant expression during development of these tumours that may involve either exclusion or inclusion of Exon 31. To test for a relationship between Exon 31 inclusion and survival, we performed Kaplan–Meier analyses grouping patients based on PSI Exon 31 (Figure 1G and Appendix AA–I). We found that in breast cancer, a higher proportion of the *RIF1-L* splice variant correlates with improved survival, indicating that across all breast cancer subtypes, a shift towards the *RIF1-S* splice variant during tumour development is associated with poor patient outcome.

### 2.2. RIF1 Splice Variant Expression Is Altered in Receptor-Positive Breast Cancers

Due to the highly heterogenous nature of cancer, tumours are classified based on genetic and molecular features. An example of this is the Prediction Analysis of Microarray 50 (PAM50) gene signature that categorises breast cancers into intrinsic molecular subtypes based on expression of a 50-gene signature [27,28,29]. This gene signature correlates with the expression of several receptors, which are the main basis for histological tumour classification. Luminal A (LumA) and Luminal B (LumB) are positive for the oestrogen (ER) and progesterone (PR) receptors; HER2-enriched (HER2E) shows overexpression of the Human Epidermal Growth Factor Receptor 2 (HER2); while the ‘basal-like’ subtype is generally ‘triple-negative’, lacking expression of all three receptors. The receptor-positive tumour subtypes are typically slower growing and are associated with better patient outcomes, partly due to the range of targeted therapies available. Basal breast cancers, in contrast, tend to be more aggressive with a higher mortality rate. To explore whether the decrease in *RIF1-L* proportion observed in breast cancer (Figure 1F) is associated with a specific cancer subtype or with all breast cancers, we subdivided patients based on their PAM50 classification and repeated the analysis of Figure 1 for the 111 patients with matched normal samples available. For LumB, HER2E, and basal cancers we observed no significant change in total *RIF1* mRNA expression between tumour and matched normal samples separated (Figure 2A), while for LumA cancers we observed a marginally significant drop in expression, consistent with there being no significant change in *RIF1* expression when considering all breast cancers combined (Figure 1C). The previous report of increased expression of RIF1 in breast tumour sections did not examine the differences between the breast cancer subtypes [7]. Considering changes in *RIF1* splice variant expression in the different breast cancer subtypes, we found that LumA, LumB, and HER2E breast cancers showed a significantly decreased proportion of *RIF1-L* transcript (Figure 2B), in line with our observation for all breast cancer subtypes combined (Figure 1F). For basal breast cancer patients, we, however, saw no significant change in the proportion of *RIF1* splice variants expressed. To assess whether inclusion of Exon 31 is linked to overall expression level of *RIF1* in the different breast cancer subtypes, we plotted PSI values for Exon 31 against total *RIF1* mRNA expression in tumour samples (Figure 2C). We observed no strong correlation between expression of *RIF1* and inclusion of Exon 31 in any of the subtypes, as indicated by low R^2^ values. Therefore, total *RIF1* mRNA levels do not appear to predict preferential expression of a specific *RIF1* isoform in breast cancer.

### 2.3. RIF1 Expression Changes in an ER+ Breast Cancer Cell Line Under Replication Stress

Rapidly dividing cancer cells are prone to replication stress, partly due to their increased requirement for DNA replication alongside derailment of normal regulatory pathways. In cultured cells, RIF1-L or RIF1-S isoform expression is connected to the ability of cells to survive DNA replication stress, with only RIF1-L competent to fully protect cells from replication stress-inducing drugs [15]. We therefore tested whether in cultured cells, Exon 31 inclusion is altered by treatments that induce replication stress, partly to understand if replication stress itself induces the shift towards Exon 31 exclusion observed in receptor-positive breast cancer. For this analysis we used MCF7 and MDA-MB-231 breast cancer cell lines to represent ER+ and basal breast cancer, respectively. We also examined the colon cancer cell line HCT116, since a shift towards *RIF1-S* transcript was observed in colon cancer, similar to that in receptor-positive breast cancer (Figure 1F).

Cells were treated for up to 72 h with the replication stress inducer Aphidicolin (APH), which inhibits activity of the major replicative DNA polymerases (Pol α, δ, and ε), to emulate the replication stress that is a key feature of cancer cells. A concentration of 1 µM was used to induce fork-stalling and potential cell cycle arrest. We assessed the relative amounts of *RIF1-L* and *RIF1-S* mRNA by competitive RT-PCR, using a primer pair that spans Exon 31 (Figure 3A, approach validated in Appendix A). Quantification of the amplified products allows measurement of the proportions of each splice variant within each sample. In all three cell lines the proportion of *RIF1-L* mRNA remained unchanged under replication stress over the 72-h period of the experiment (Figure 3B,C). We also measured the representation of total *RIF1* mRNA by RT-qPCR, quantified relative to the housekeeping gene *GAPDH*. This analysis revealed a significant decrease in *RIF1* expression after both 48 and 72 hr of APH treatment in the MCF7 cell line. We saw a similar, though somewhat less marked, effect in HCT116 cells, with total *RIF1* mRNA expression only significantly altered after 72 hrs. We observed no such change in the MDA-MB-231 cell line (Figure 3B). This may indicate a difference in response to replication stress in the ER+ cancer MCF7 cell line compared to the basal breast cancer MDA-MB-231 cell line, occurring without an effect on the proportions of splice variants expressed.

### 2.4. Length-Biased Exon Exclusion in Receptor-Positive Breast Cancer Subtypes

While this work was in progress, it was reported that splicing dysregulation occurs across many cancer types [30]. The splicing changes observed showed an apparent length dependence, with a tendency for exons that are excluded in cancer to be short in breast and other cancers. Considering that *RIF1* Exon 31 is itself relatively short at 78 nucleotides, we sought to extend our analysis by comparing the observed changes in *RIF1* splicing to general splicing changes occurring in the different breast cancer subtypes. Our aim was to understand whether the observed changes in *RIF1* Exon 31 inclusion in the different subtypes reflect a general difference between receptor-positive and receptor-negative breast cancers in terms of effects on short and long exon inclusion.

For this analysis we first extracted from TCGA SpliceSeq the PSI values for all observed exon skips (excluding instances of multiple exon skips for simplicity). For each of the exon skips, we then calculated an average PSI value (PSI_av_) across all patients (Figure 4A). For each exon skip, a ‘ΔPSI_av_’ value was then obtained by subtracting the PSI_av_ in normal cells from the PSI_av_ in tumours to obtain a value expressing how much the inclusion of an exon changes on average in each breast tumour type. Figure 4A (right) shows a histogram of ΔPSI_av_ values for all the exons in the analysis. The sharp peak at 0 illustrates that for most alternatively spliced exons, their rate of inclusion is unchanged in cancer. However, tails on either side of the peak show that some exons do tend to be excluded more frequently in cancer (negative ΔPSI_av_ values, on the left of the histogram) while others tend to be included more frequently (positive ΔPSI_av_ values, on the right of the histogram). To focus analysis on splicing changes in reasonable magnitude, we selected for further analysis only those exon-skip events whose ΔPSI_av_ changed by at least 0.1 between normal and tumour tissue (i.e., with values outside the vertical black lines in the histogram), similar to the approach used by Zhang et al. (2022) [30] to designate Cancer-Altered exons. These Cancer-Altered exons were then further subdivided according to length by designating exons of length less than 80 nt as ‘Short’ and those greater than 80 nt as ’Long’. Our threshold of 80 nt to designate exons as Short is slightly longer than the 60 nt threshold used by Zhang et al. (2022) [30], but still considerably less than the average length of human exons of 150 nt. Results were very similar when taking 80 or 60 nt as the cutoff threshold. The ΔPSI_av_ values were then examined in the four different breast cancer types separately, plotting histograms to show the number of Cancer-Altered Short exons that show preferential inclusion (positive ΔPSI_av_ values) or exclusion (ΔPSI_av_ values) in cancer (Figure 4B, with the Cancer-Altered Short exon cohort shown in yellow). We likewise plotted the number of Cancer-Altered Long exons that show preferential inclusion or exclusion (Figure 4B, Cancer-Altered Long exon cohort shown in blue).

Examining these plots reveals that, in LumA and LumB breast cancer, a higher number of Cancer-Altered Short exons are excluded rather than included, as indicated in Figure 4B by the fact that the majority of Short exons have negative ΔPSI values. Specifically, the yellow histograms show that for LumA and LumB breast cancers, respectively, 70% (274 out of 390) and 65% (350 out of 542) of Cancer-Altered Short exons have ΔPSI values < 0.1. Cancer-Altered Long exons were included or excluded at similar rates (i.e., blue histogram plots for LumA and LumB cancers show a similar number of exons with values above 0.1 or below −0.1). In HER2E cancer, 67% of Cancer-Altered Short exons were excluded in cancer, a similar proportion as seen for LumA and LumB cancers. In HER2E cancer, Cancer-Altered Long exons also show a slight tendency to be excluded rather than included, although at 55% excluded, the tendency is less marked than for Short exons. In basal breast cancer, however, we observed that all exons, regardless of length, are more often excluded in cancer. Specifically, 66% (362 out of 552) of Cancer-Altered Short exons and 62% (608 out of 986) of Cancer-Altered Long exons shifted towards exclusion in basal breast cancer. Therefore, basal breast cancer may not share the tendency of LumA, LumB, and HER2E cancers to favour Short exons when designating which exons will be preferentially excluded.

To assess the significance of these splicing changes, for each cancer type we plotted PSI_av_ values for Cancer-Altered exon skip events (i.e., those with a ΔPSI_av_ above 0.1 or below −0.1), separately for Cancer-Altered Long and Cancer-Altered Short exons, and performed unpaired Wilcoxon rank-sum tests (Figure 4C). For LumA, LumB, and HER2E cancer subtypes, Cancer-Altered Short exons showed a significant tendency to be excluded rather than included in tumour tissue when compared to matched normal tissue (compare solid yellow with hatched yellow box plots in each graph). For the same three cancer subtypes, Cancer-Altered Long exons were equally likely to show increased inclusion or exclusion (Figure 4C, compare solid blue with hatched blue box plots). However, in basal breast cancer, the shift towards exclusion was significant not only for Cancer-Altered Short but also for Cancer-Altered Long exons. This result may suggest that while receptor-positive (LumA, LumB, and HER2E) are able to regulate which exons will be preferentially excluded during carcinogenesis (favouring shorter exons for exclusion), basal breast cancer may lack such control and instead shift towards exon exclusion, regardless of exon length.

To investigate any functional associations of the alternative splicing dysregulation observed in different breast cancer subtypes, genes containing Cancer-Altered Long and Short exons were subjected to Gene Ontology (GO) analysis (Appendix A). The different breast cancer subtypes were similar in enriched GO processes, with the majority of the top 10 enriched processes associated with cytoskeletal organisation for all subtypes. This suggests that the genes undergoing differential alternative splicing in cancer are likely contributors to key cellular processes, such as altered cellular motility and cell–cell adhesion dynamics, which are functionally relevant for cancer progression and metastasis.

### 2.5. Changes in RIF1 Splicing Are Typical for Receptor-Positive but Not Basal Breast Cancer Subtypes

We considered how well the observed changes in *RIF1* splicing (Figure 2B) conform to these general patterns for the four cancer subtypes. For LumA, LumB, and HER2E cancers, *RIF1* Exon 31 is within the grouping of Cancer-Altered Short exons showing increased exclusion, given that it has a ΔPSI_av_ less than −0.1 in each of these cancer subtypes (−0.132, −0.155, and −0.161, respectively). Our finding that LumA, LumB, and HER2E cancers shift towards expressing a higher proportion of *RIF1-S* (Figure 2B) is therefore consistent with the general bias for Short exons to be skipped in LumA, LumB, and HER2E tumours. In other words, *RIF1* Exon 31 is behaving ‘typically’ for Cancer-Altered Short exons in being more often excluded in these receptor-positive cancer subtypes (Figure 2B). However, in basal breast cancer, the situation for *RIF1* Exon 31 is different. In the basal breast cancer subtype, Cancer-Altered Short and Cancer-Altered Long exons show a similar tendency to be excluded. However, for basal breast cancer, *RIF1* Exon 31 is not within the cohort of Cancer-Altered Short exons, because with a ΔPSI_av_ value of 0.02 (Figure 2B), its level of inclusion is only marginally changed in basal cancer compared to normal tissue (since its ΔPSI_av_ lies between −0.1 and 0.1). *RIF1* Exon 31 is therefore behaving unusually in basal breast cancer in not showing a significant shift towards exon exclusion. The fact that *RIF1* does not appear to alter its splicing pattern in basal breast cancer is surprising, particularly in comparison to its behaviour in LumA, LumB, and HER2E cancer subtypes, where it behaves as a ‘typical’ Cancer-Altered Short exon in its tendency to show increased exclusion in these receptor-positive cancers.

## 3. Discussion

Splicing dysregulation in cancer has been well established and is regarded as one of the drivers of cancer initiation and progression [19,31]. Here, we investigate changes in expression level and splicing of transcripts encoding RIF1, an important genome stability regulator that coordinates DNA replication and repair, whose expression has been reported to be derailed in cancer [5,6,7,8]. We find that overall *RIF1* mRNA expression levels are increased in colon and lung tumours and decreased in thyroid tumours when compared to matched normal tissue controls from the same patients (Figure 1C). Analysis of alternatively spliced transcripts in patient-matched normal and cancer samples revealed a shift towards exclusion of *RIF1* Exon 31 and expression of the *RIF1-S* splice variant in several cancers (Figure 1F), including breast, colon, lung, liver, and prostate cancer.

Analysis of general changes in alternative splicing occurring in breast cancer subtypes revealed that receptor-positive breast cancers (LumA, LumB, and HER2E) show a tendency to specifically exclude Short exons. The observed shift towards expression of *RIF1-S* in these breast cancer subtypes (Figure 2B) therefore conforms with transcript processing changes that are generally observed to occur in these cancer subtypes. A shift towards exon exclusion also occurs in receptor-negative (basal) breast cancer, but this cancer subtype shows no preference for exclusion of Short exons, instead excluding Long and Short exons at similar rates (Figure 4B,C). Interestingly, in basal breast cancer the *RIF1* transcript did not follow the ‘typical’ trend. Rather, basal breast cancer showed no change in *RIF1* splice variant expression when comparing matched normal and tumour samples, continuing to express the *RIF1* variants in proportions similar to those in normal tissue (Figure 2B).

Differences in global alternative splicing between the breast cancer subtypes have been described previously [32,33]. Clustering analysis based on splicing patterns can differentiate the cancer subtypes, with a particular distinction observed between receptor-positive and negative tumours [34,35,36]. Our observation of different tendencies for breast cancer subtypes to exclude short exons is likely to contribute to this effect. In terms of the mechanism of this length-dependent splicing alteration, Zhang et al. (2022) [30] report that cancer-associated exon exclusion is associated with elevated speed of transcription by RNAPII, with short exons particularly sensitive to RNAPII kinetics. Zhang et al. also show that the mis-splicing occurs more frequently in genes functionally relevant to cancer. Such genes are likely to be subject to enhanced rates of transcription in tumours, potentially explaining their propensity to exhibit exon skipping. Transcriptional profiles do vary between the breast cancer subtypes [28,37], although there is no current evidence to suggest differences in transcriptional speed.

We identified an increase in overall *RIF1* mRNA expression in colon and lung tumours compared to matched normal tissue controls from the same patients (Figure 1C). Of these, *RIF1* expression has only been previously studied in NSCLC, the most common lung cancer subtype [24]. The earlier analysis performed a chemosensitivity study, where tumour samples were collected from 89 patients before chemotherapy, then RIF1 expression was quantified and correlated with their response to the drug. Higher RIF1 expression was significantly correlated with poorer response to platinum-based chemotherapy. For breast cancer, despite the association of RIF1 with various pro-oncogenic processes, we observed no significant change in *RIF1* gene expression between normal and cancer tissue (Figure 1C) and no association of *RIF1* mRNA expression level with survival outcomes (Figure 1D). While TCGA contains information from a large number of patients, all samples are treatment naïve, and there is limited data available regarding therapeutic interventions and treatment responses. Therefore, a breast cancer chemosensitivity study, similar to the one undertaken for NSCLC, would be useful for understanding the relationship between RIF1 expression and tumour response to chemotherapy.

Our investigation of the alternative splicing of RIF1 in cancer identified a shift towards the *RIF1-S* splice variant in various cancer types (Figure 1F). We find this same tendency to exclude *RIF1* Exon 31 in LumA, LumB, and HER2E breast cancers, although not in basal breast cancer (Figure 2B). This shift towards *RIF1-S* expression is perhaps surprising given that only the RIF1-L protein variant is capable of protecting cells from DNA replication stress, with RIF1-L conferring a survival advantage on replication-stressed cells cultured in vitro [15] by forming an interaction with BRCA1 that promotes replication fork recovery [17]. Given that cancer cells must cope with high levels of inherent replication stress (and in patients are frequently subjected to replication-stress-inducing chemotherapeutic drugs), it is perhaps surprising that the *RIF1-L* isoform tends to be downregulated in cancer cells. However, it is also known that replication stress and associated genomic instability are important contributors to cancer initiation and progression, and as such are among the ‘hallmarks of cancer’ [31]. Therefore, the shift towards RIF1-S expression and resulting increased replication stress may promote further metastatic transformation over the timescales of tumour development.

For basal breast cancer, the RNA processing pattern of the *RIF1* transcript was unexpected. Although basal breast cancer shares the tendency of receptor-positive subtypes to show increased exon exclusion, Long and Short exons are equally likely to be excluded (Figure 4B,C). Therefore, the retention of *RIF1* Exon 31 in this tumour type (Figure 2B) contrasts with its behaviour in the receptor-positive breast cancers, where it tends to be excluded along with other Cancer-Altered Short exons. Basal cancers are the most aggressive form of breast cancer, associated with a poorer response to chemotherapy. One intriguing possibility is that a higher proportion of *RIF1-L* in this cancer subtype may contribute to malignancy by conferring a survival advantage to the tumour cells, particularly in the context of replication stress. Such a survival advantage might reflect the specific ability of RIF1-L to interact with BRCA1, which promotes recombination-mediated recovery of broken replication forks, which arise under the prolonged stress conditions characteristic of cancer cells [17].

Collectively our data highlight a propensity for receptor-positive breast tumours to exclude Short exons, including *RIF1* Exon 31, with evidence that this aberrant splicing is associated with the pathogenesis of the disease and ongoing prognosis. While several studies have reported changes in RIF1 expression levels in cancer or in cancer cell lines, to our knowledge this is the first study to examine *RIF1* splicing in stratified cancer subtypes or indeed to report on length-based exon splicing patterns in breast cancer subtypes. Future work will determine roles and consequences of RIF1 splice variant expression changes in these tumour types.

## 4. Materials and Methods

### 4.1. Analysis of RIF1 Exon 31 Inclusion

Percent spliced-in (PSI) values for Exon 31 of *RIF1* were downloaded from TCGA SpliceSeq for both normal and cancer tissue for all cancer types (bioinformatics.mdanderson.org/TCGASpliceSeq) [26]. Patients with no ‘matched normal’ tissue data available were excluded, and tumour types with fewer than 10 remaining patients were excluded. *RIF1* gene expression (as log2(norm count + 1) from TCGA) was obtained from UCSC Xena (xena.ucsc.edu) [38], and clinical data were obtained from cBioPortal for Cancer Genomics (cbioportal.org) [39] and aligned with PSI values using unique patient identifiers. For analysis of expression in specific breast cancer subtypes, patients without available PAM50 subtypes were excluded. Claudin-low (CLOW) and normal-like subtypes were excluded due to low patient numbers. Statistical comparison of *RIF1* mRNA and PSI Exon 31 values in normal and tumour samples was performed in R version 4.1.2 with a paired Wilcoxon signed-rank test using ggplot2 version 3.4.1. Linear regression analyses and calculation of R-squared (R^2^) correlation coefficients of *RIF1* mRNA against PSI Exon 31 values were performed in R using ggplot2.

### 4.2. Analysis of Global Exon Skip Events

PSI values for all observed exon skip events (including novel events), along with exon length information, were downloaded from TCGASpliceSeq for all breast cancer samples (normal and cancer). Filters for splice events were set at a minimum of 75% of samples having PSI value available. Breast cancer tumour samples with matched normal samples were selected, and those without PAM50 subtype available were excluded. Claudin-low (CLOW) and normal-like subtypes were excluded due to low patient numbers. Exon skips involving multiple exons were excluded. An average PSI value (PSI_av_) was calculated for each exon skip event, separately for normal and tumour tissue. ΔPSI_av_ was calculated by subtracting the PSI_av_ of the normal samples from the PSI_av_ of the tumour samples. To identify exons altered in cancer (Cancer-Altered exons), exon skip events with ΔPSI_av_ values between −0.1 and 0.1 were excluded. Statistical comparison of PSI values between tumour and normal tissue was performed in R with the unpaired Wilcoxon rank-sum test (Mann–Whitney test) using ggplot2.

### 4.3. Survival Analysis

Survival analysis on datasets compiled as described above was carried out using KM Plotter (kmplot.com) [40,41]. Cohorts of patients were split by median expression values of *RIF1*, or median PSI *RIF1* Exon 31. KM plots using progression-free survival (months) were generated, including number at risk, hazard ratio (HR), 95% confidence interval, and log-rank *p*-values. HR was calculated using univariate Cox regression.

### 4.4. Gene Ontology Analysis

Genes that contained Cancer-Altered Long and Short exons in each breast cancer subtype were used for GO analysis using WebGestalt (webgestalt.org) [42]. The top 10 enriched cellular processes were generated using over-representation analysis.

### 4.5. Cell Culture

MDA-MB-231 and MCF7 cells, gifted by Prof. Valerie Speirs, were cultured in RPMI medium supplemented with 10% foetal bovine serum at 37 °C with 5% CO_2_. HCT116 cells were cultured in McCoy’s 5A medium supplemented with 10% foetal bovine serum and 2 mM L-glutamine at 37 °C with 5% CO_2_.

### 4.6. Analysis of Total RIF1 mRNA and Splice Variant Expression in Vitro

Cells were treated with 1 µM of Aphidicolin (APH) (Sigma-Aldrich, St. Louis, MO, USA) for the indicated time periods in the normal culture conditions indicated above. RNA was harvested using the Monarch Total RNA Miniprep Kit (NEB, Ipswich, MA, USA) according to the manufacturer’s instructions. RNA was reverse transcribed using the SuperScript IV First-Strand Synthesis Kit (Invitrogen, Carlsbad, CA, USA) according to the manufacturer’s instructions. *RIF1* splice variant transcripts were amplified by competitive PCR using a primer pair spanning Exon 31 (Appendix A, Figure 3A) using Phire Green Hot Start II PCR Master Mix (Thermo Fisher Scientific, Waltham, MA, USA), as described in Appendix A. After separation on 2% agarose gel, bands were visualised using ethidium bromide or SYBR Green I dye and imaged using the Bio-Rad ChemiDoc Touch system (Bio-Rad, Hercules, CA, USA). Relative abundance of splice variants, indicated by band intensity, was quantified using ImageLab version 6.1 (Bio-Rad), as described in Appendix A. Statistical comparison was performed with paired, one-tailed *t*-tests in R using ggplot2. Total *RIF1* mRNA was quantified using PrimeTime™ Predesigned qPCR Probe Assays (Integrated DNA Technologies, Coralville, IA, USA) (Appendix A) in a multiplex RT-qPCR reaction on a LightCycler 480 II system (Roche Diagnostics, Basel, Switzerland. Absolute quantification was carried out using an on-plate standard curve and normalised to *GAPDH* using the ΔΔCt method. Statistical comparison was performed with paired, two-tailed *t*-tests in R using ggplot2.

## Figures and Tables

**Figure 2 ijms-26-07308-f002:**
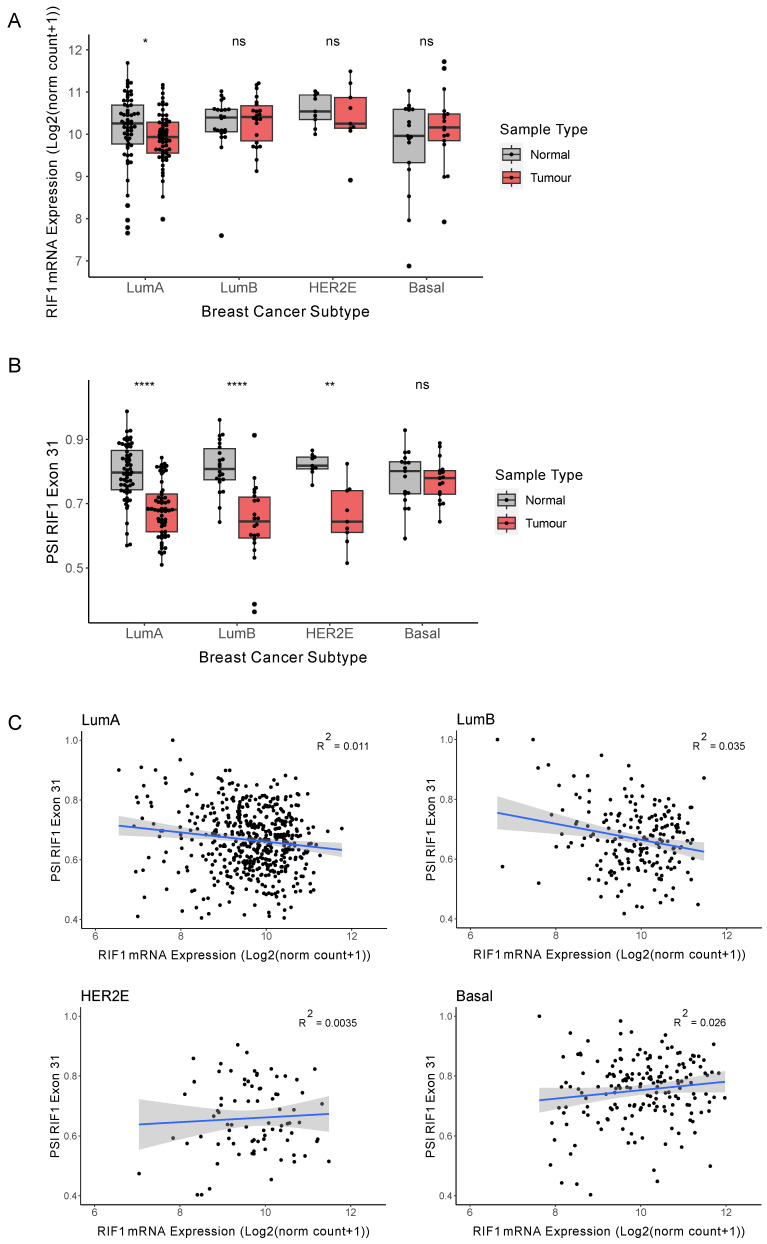
The *RIF1-L* proportion is decreased in receptor-positive breast cancers. (**A**) *RIF1* mRNA expression (as log2(norm count + 1)) in tumour (red) and matched normal samples (grey) in different breast cancer subtypes. Statistical comparison was carried out using paired Wilcoxon signed-rank test, * *p* < 0.05, ns: non-significant. (**B**) PSI values for *RIF1* Exon 31 in tumour (red) and matched normal samples (grey) in different breast cancer subtypes. Statistical comparison was carried out using paired Wilcoxon signed-rank test, ** *p* < 0.01, **** *p* < 0.0001, ns: non-significant. (**C**) PSI values for *RIF1* Exon 31 plotted against *RIF1* mRNA for each tumour sample in different breast cancer subtypes. The blue line shows linear regression analysis, and the shaded area indicates 95% confidence intervals, with R^2^ values located on each plot.

**Figure 3 ijms-26-07308-f003:**
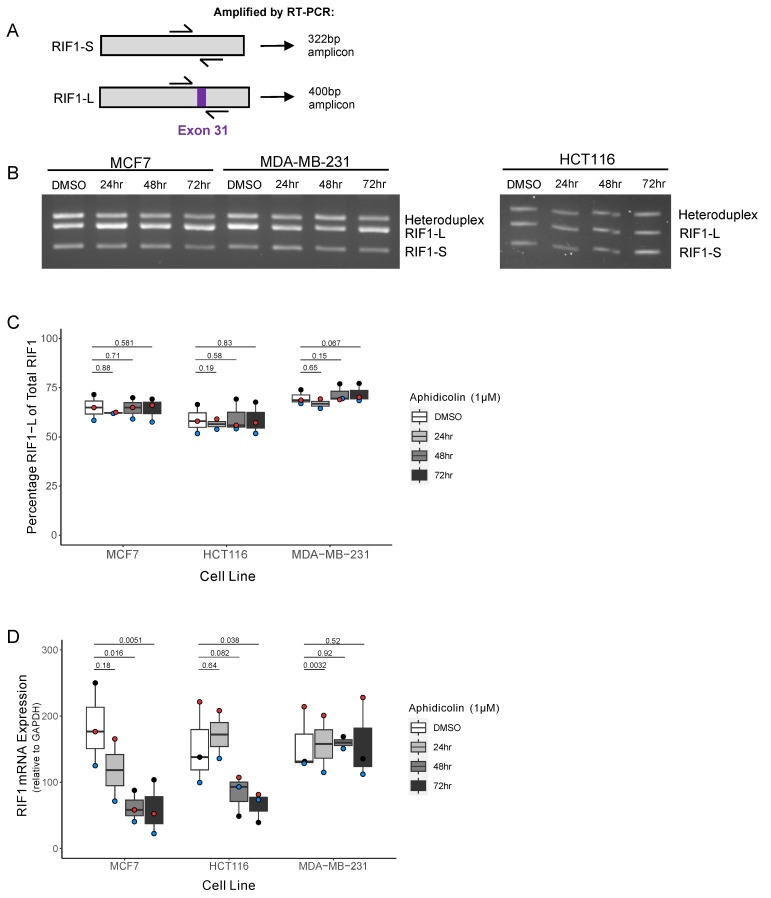
The ER+ breast cancer cell line shows decreased *RIF1* expression but no splice variant shift under drug-induced replication stress. (**A**) Schematic diagram showing approximate binding location of primer pair used to distinguish *RIF1-S* and *RIF1-L*. Amplification of *RIF1-S* mRNA by RT-PCR produces a 322 bp product, while *RIF1-L* mRNA produces a 400 bp product. (**B**) Representative image of *RIF1* splice variant visualisation after agarose gel separation, following RT-PCR as described in (**A**). MCF7, MDA-MB-231, and HCT116 cell lines were treated with 1 µM Aphidicolin (APH) for the indicated time periods. (**C**) Quantification of analysis in (**B**) (as described in Appendix A), showing percentage *RIF1-L* of total *RIF1* in MCF7, HCT116, and MDA-MB-231 cell lines, treated with 1 µM Aphidicolin (APH) for indicated time periods. The plot shows the median and range from three biological replicates, with each replicate distinguished by coloured points. Statistical significance was calculated using paired single-tailed *t*-tests. (**D**) *RIF1* mRNA expression determined by RT-qPCR in MCF7, HCT116, and MDA-MB-231 cell lines, treated with 1 µM Aphidicolin (APH) for indicated time periods. Absolute quantification values were normalised to *GAPDH*. The plot shows the median and range from three biological replicates (two replicates for the 24 h timepoint), with each replicate distinguished by coloured points. Statistical significance was calculated using paired two-tailed *t*-tests.

**Figure 4 ijms-26-07308-f004:**
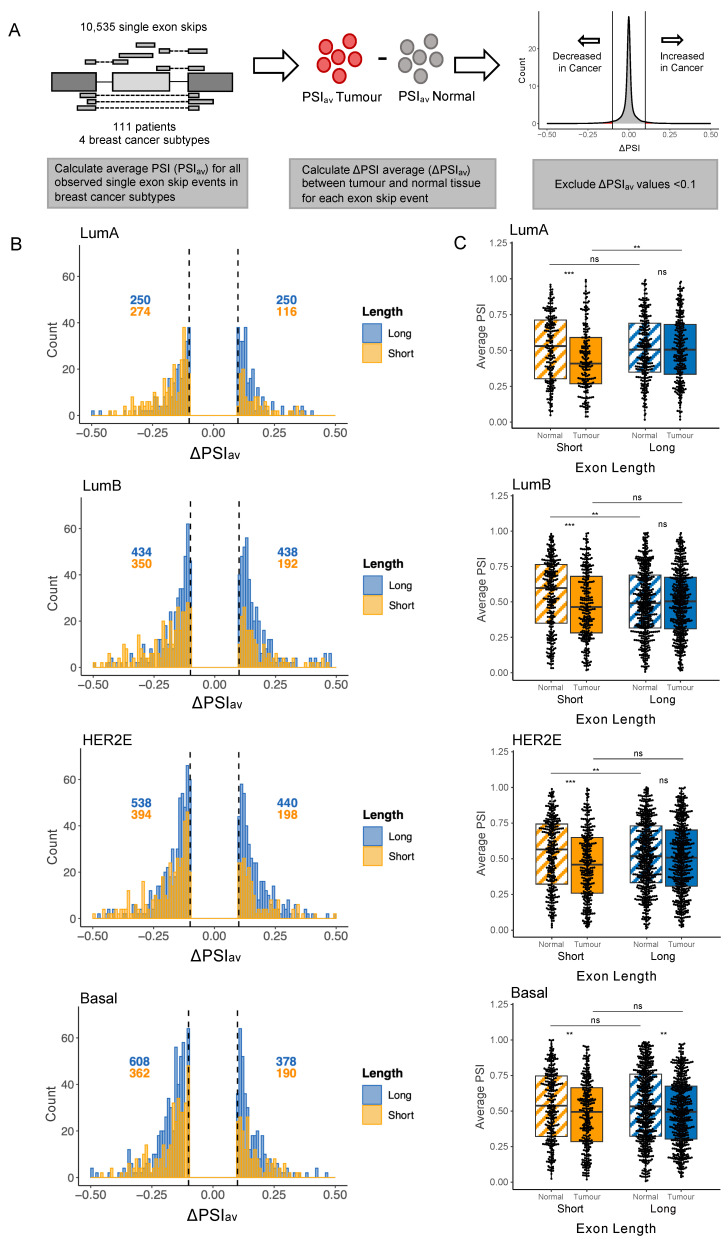
Short exons are preferentially excluded in receptor-positive breast cancer. (**A**) Process of calculating the difference in average PSI value between normal and tumour tissue for all exon skip events. Using data downloaded from TCGA SpliceSeq, an average PSI value (PSI_av_) was calculated for each exon skip in normal and tumour samples for all 111 breast cancer patients with both normal and tumour samples available. Average PSI in the normal sample was subtracted from average PSI in the tumour to obtain ΔPSI_av_. Exons with a PSI_av_ value between −0.1 and 0.1 were excluded to focus on those showing changes between normal and tumour tissue. (**B**) Histograms show number and distribution of ΔPSI values for exons with ΔPSI_av_ values less than −0.1 or greater than 0.1. Patients were separated by breast cancer subtype. Exons are divided into Long (>80 nt, plotted in blue) and Short (≤80 nt, plotted in yellow) cohorts. Numbers of exons with ΔPSI_av_ values falling above 0.1 or below −0.1 are annotated in blue (for Long exons) and yellow (for Short exons). (**C**) At the left of each graph are plotted the PSI_av_ values for Short exons in normal samples (hatched yellow) and tumour samples (solid yellow), for Cancer-Altered exons, i.e., those with ΔPSI_av_ values less than −0.1 or greater than 0.1. At right are plotted the PSI_av_ values for Long exons in normal samples (hatched blue) and tumour samples (solid blue), also for Cancer-Altered exons. The four plots from top to bottom show patients grouped by breast cancer subtype. Statistical comparison was carried out using unpaired Wilcoxon rank-sum test, ** *p* < 0.01, *** *p* < 0.001, ns: non-significant.

## Data Availability

The datasets analysed during this study are available from TCGA SpliceSeq (bioinformatics.mdanderson.org/TCGASpliceSeq), UCSC Xena (xena.ucsc.edu), and cBioPortal for Cancer Genomics (cbioportal.org).

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
