# Peer review of "Dysregulated Alternative Splicing in Breast Cancer Subtypes of RIF1 and Other Transcripts"

_ijms, 2025, doi:10.3390/ijms26157308_

Round 1
Reviewer 1 Report
Comments and Suggestions for Authors
The authors examined altered RIF1 mRNA splicing in breast cancer by public database and in bitro analysis. Although some data is interesting, there are some critical points to be addressed.
1. The present manuscript lack the comparison of total RIF1 mRNA expression or RIF1-exon 31 among the subtype of breast cancer, making it difficult to discuss the differential clinical significance of RIF1 splicing.
2. Regarding Figure 1D, there seems to be opposite descriptions.
132-134: However, in other cancer types, including breast cancer, we did not find a significant correlation 133 between RIF1 expression and survival in Breast cancer (Fig. 1D & S1).
386-388: For breast cancer, despite there being no significant change in RIF1 gene expression between normal and cancer tissue, survival data analysis shows that patients with high tumour RIF1 expression are more likely to have worse survival outcomes (Fig. 1D).
3. The authors performed competitive RT-PCR to address the change of RIF1 mRNA splicing. Is this method really suitable for mRNA splicing analysis? Usually, competitors are added into the PCR reaction and mRNA expression is quantified based on competitor-dose dependent inhibition of PCR reaction. The authors have not provided the electrophoresis images showing PCR inhibition by competitors. In addition, two different PCR products are generated by common PCR primers. There is a concern that the two amplification products may interfere with each other's amplification, affecting the quantitative nature of competitive PCR. It is recommended to analyze by real time PCR.
4. Discission section is too short.
Reviewer 2 Report
Comments and Suggestions for Authors
The mauscript is well-written. Yet, the figures have to be improved. COncerning the introduction it is recommened to have more elaboration on the mechanisms of dysregulation for the "length-dependent splicing dysregulation" in some breast cancer subtypes.
The manuscript states that RIF1-Long protects against replication stress. this has to be discussed in more details in the discussion
the statistical methods used to establish correlations and significance in TCGA data analysis must be clearly mentioned in the manuscript
Reviewer 3 Report
Comments and Suggestions for Authors
Parker and colleagues present a paper on the differences in splicing of the RIF1 gene in breast cancer subtypes. Given the importance of RIF1 in the DNA damage response, this research is of high importance; however, there are a couple of concerns that the authors must address before publication.
Major points
The presented idea heavily relies on the experimental verification of the in silico findings regarding RIF1 expression and alternative splicing. Although the changes, or lack thereof, in alternative splicing are supported by densitometric analysis (Figure 3A) of the endpoint PCR shown in Figure S3. Besides the fact that Figure S3 should be part of the main text, given its importance, the gel is presented without proper controls (a no-template negative control and a housekeeping gene amplicon as a positive one are expected), and there is no mention of the amplification conditions, rising suspicion about the reproducibility of the experiment. The choice of performing a densitometric analysis on gel bands seems odd, again, considering the importance of the experiment. It is well known that endpoint PCR amplification eventually plateaus, making it virtually impossible to discern between amplifications of similar quantities of source material. The RIF1-L transcript quantity in each cell line must be evaluated by qPCR. I suggest using a reverse primer within exon 31.
The Discussion section must be improved. First, it largely re-states the results; and second, it contains no references for the interesting ideas put forward. For instance, the authors suggest that “a higher proportion of RIF1-L in this cancer subtype may contribute to malignancy” (Lines 415-416) without elaborating on a possible mechanism.
Minor points
The title is somewhat misleading, as it suggests that multiple genes were analyzed, while only RIF1 was analyzed in depth.
It is unclear what downstream effects the authors are referring to in lines 45-47.
Figure 3 lacks significance markers.
Figure 3B depicts a relative mRNA quantification, as described in the Methods section, yet line 228 mentions absolute quantification values. Please clarify.
Round 2
Reviewer 1 Report
Comments and Suggestions for Authors
The manuscript was well revised.
Reviewer 3 Report
Comments and Suggestions for Authors
The authors have addressed the concerns raised by all reviewers satisfactorily